# A Mediterranean *Alexandrium taylorii* (Dinophyceae) Strain Produces Goniodomin A and Lytic Compounds but Not Paralytic Shellfish Toxins

**DOI:** 10.3390/toxins12090564

**Published:** 2020-09-01

**Authors:** Urban Tillmann, Bernd Krock, Stephan Wietkamp, Alfred Beran

**Affiliations:** 1Alfred Wegener Institute-Helmholtz Zentrum für Polar- und Meeresforschung, Am Handelshafen 12, D-27570 Bremerhaven, Germany; bernd.krock@awi.de (B.K.); Stephan.Wietkamp@awi.de (S.W.); 2National Institute of Oceanography and Applied Geophysics—OGS, via Piccard 54, I-34151 Trieste, Italy; aberan@inogs.it

**Keywords:** goniodomin, *Gessnerium*, toxins, paralytic shellfish poisoning (PSP), spirolides, lytic compounds

## Abstract

Species of the dinophyte genus *Alexandrium* are widely distributed and are notorious bloom formers and producers of various potent phycotoxins. The species *Alexandrium taylorii* is known to form recurrent and dense blooms in the Mediterranean, but its toxin production potential is poorly studied. Here we investigated toxin production potential of a Mediterranean *A. taylorii* clonal strain by combining state-of-the-art screening for various toxins known to be produced within *Alexandrium* with a sound morphological and molecular designation of the studied strain. As shown by a detailed thecal plate analysis, morphology of the *A. taylorii* strain AY7T from the Adriatic Sea conformed with the original species description. Moreover, newly obtained Large Subunit (LSU) and Internal Transcribed Spacers (ITS) rDNA sequences perfectly matched with the majority of other Mediterranean *A. taylorii* strains from the databases. Based on both ion pair chromatography coupled to post-column derivatization and fluorescence detection (LC-FLD) and liquid chromatography coupled to tandem mass spectrometry (LC-MS/MS) analysis it is shown that *A. taylorii* AY7T does not produce paralytic shellfish toxins (PST) above a detection limit of ca. 1 fg cell^−1^, and also lacks any traces of spirolides and gymnodimines. The strain caused cell lysis of protistan species due to poorly characterized lytic compounds, with a density of 185 cells mL^−1^ causing 50% cell lysis of cryptophyte bioassay target cells (EC_50_). As shown here for the first time *A. taylorii* AY7T produced goniodomin A (GDA) at a cellular level of 11.7 pg cell^−1^. This first report of goniodomin (GD) production of *A. taylorii* supports the close evolutionary relationship of *A. taylorii* to other identified GD-producing *Alexandrium* species. As GD have been causatively linked to fish kills, future studies of Mediterranean *A. taylorii* blooms should include analysis of GD and should draw attention to potential links to fish kills or other environmental damage.

## 1. Introduction

Exceptional densities of marine microalgae, commonly reported as blooms, are recurrently observed in many coastal areas around the world. A number of dinophycean microalgae are producers of potent phycotoxins which, during such blooms, may have major ecological (e.g., fish kills), economic (e.g., on tourism or exploitation of marine resources) and/or sanitary impacts (e.g., human poisoning). Among toxigenic dinophytes, the genus *Alexandrium* Halim is perhaps the most intensely studied group. The taxonomic history of this typical gonyaulacoid genus is quite complex and includes numerous rearrangements of species formerly classified in *Gonyaulax*, *Protogonyaulax*, *Gessnerium*, *Goniodoma* and *Pyrodinium* [1,2]. Some new species have recently been described and today the genus *Alexandrium* comprises 34 species. The importance of the genus is mainly attributed to the devastating effects of toxigenic blooms related to human poisoning via contaminated shellfish. Species of *Alexandrium* may produce a large variety of toxic compounds including paralytic shellfish toxins (PST) (saxitoxin and derivatives), spiroimines (spirolides, gymnodimines), goniodomins (GD), and poorly characterized extracellular lytic compounds [3]. Among these various compounds, the neurotoxic saxitoxin and its derivatives are the most well-known and widely distributed, and blooms of PST producing species regularly have devastating effects on aquaculture industry around the world. For example, in 2016 a severe *Alexandrium catenella* bloom of outstanding intensity and geographical extent hit Chile with devastating effects on salmon aquaculture [4,5].

Whereas toxin production is well studied for the main PST-producing species, for example for species of the former *tamarense*/*fundyense*/*catenella* species complex or for *A. minutum*, much less is known about toxin production potential of other *Alexandrium* species. One of these is *Alexandrium taylorii*. The species was described by Balech [6] in the French Atlantic (Bay of Arcachon, France) and since then has been reported from various Mediterranean areas [7] as well as from Indonesia [8], Malaysia [9], and Japan [10]. *Alexandrium taylorii* is a high biomass producer species causing very dense and recurrent blooms in various parts of the Mediterranean Sea including the Catalano-Balearic, Adriatic, Tyrrhenian and Ionian Sea, where peak densities of 10^6^–10^7^ cells L^−1^ and intense water discolorations are reported [7,11,12,13,14,15,16]. However, toxin production potential of *A. taylorii* is poorly known and there are only few and partly contradictory studies available. Mediterranean strains are usually listed and cited as non-PST-producers [15,17,18], however, this belief is not based on actual data or is simply based on “pers. comm.” information [19]. The same refers to a strain classified as *A. taylorii* from Indonesia, which was referred to as a strain that did not produce PST, but again only based on “pers.comm.” and not on published data [8]. Nevertheless, for one Mediterranean strain, AY1T, methodological details confirming lack of detectable PST was published [20]. In contrast, PST production based on high performance liquid chromatography (HPLC) toxin analysis was claimed for a Malaysian strain of *A. taylorii* [9].

Moreover, *A. taylorii* has been reported to severely affect oyster larvae [8] and to produce hemolytic exotoxins [10]. It must be noted, however, that in both reports neither morphological nor sequence data is provided supporting the species identification. On the other hand, the Mediterranean strain AY1T, for which sequence data are available in GenBank, was shown to immobilize and lyse a protistan grazer which is indicative of the production of extracellular lytic compounds by *A. taylorii* [20].

Morphological evidence, i.e., a pentagonal first apical plate disconnected from the apical pore plate [6], indicates a close relationship of *A. taylorii* with other species of the subgenus *Gessnerium* as defined by Balech [21], and such a relationship is confirmed in phylogenetic trees for *A. taylorii* with *A. monilatum*, *A. pseudogonyaulax*, *A. hiranoi*, and *A. satoanum* [22,23]. Interestingly, species of this cluster are known as producers of goniodomin A (GDA) (Figure 1), a potent antifungal toxin associated with invertebrate mortality [24], which was first identified by Sharma et al. [25]. Whereas the species identity of the *Alexandrium* sp. source organism studied by Sharma et al. [25] cannot be determined retrospectively [26], GDA has been identified in *A. hiranoi* [27], *A. monilatum* [28], and *A. pseudogonyaulax* [29]. The other two species of the phylogenetic cluster, *A. satoanum* and *A. taylorii*, have never been tested for the presence of this toxin.

The aim of the present study is thus to investigate the toxin production potential of a Mediterranean *A. taylorii* strain by combining state-of-the-art screening for various toxins known to be produced within *Alexandrium* with a sound morphological and molecular designation of the studied strain.

## 2. Results

### 2.1. Species Identification

Cells were slightly variable in shape from subspherical to irregularly hexagonal (Figure 2A,B) without significant dorsoventral compression. The epitheca was rounder than the trapezoidal hypotheca. Cell size ranged from 34.3 to 48.2 µm in cell length (mean 40.4 ± 3.3 µm) and 33.6 to 49.3 µm in cell width (mean: 41.6 ± 3.7 µm) with a mean length/width ratio of 0.97 ± 0.03 (*n* = 52). The cingulum was narrow, excavated, without lists, and ventrally displaced by slightly more than one cingular width (Figure 2C). The cell content was brownish (Figure 2C–E) and could be quite dark and granular. There were numerous regularly distributed small chloroplasts visible in fluorescence microscopy (Figure 2K,L). Position and shape of the nucleus was difficult to resolve in unstained cells, but with DAPI staining it was seen to be elongated and located in the cingular plane (Figure 2K–M) with its U-shape clearly visible in apical view (Figure 2N,O). In the culture there were two types of cell division. Cells divided in the motile stage with an oblique fission line by desmoschisis, i.e., the thecal plates were shared between the two new cells (Figure 2D,E). Additionally, cells could shed their theca (ecdysis) (Figure 2F,G) forming temporary cysts, which subsequently may undergo cell division (Figure 2H–J).

The theca was composed of thin and smooth plates which were irregularly covered by minute pores (Figure 3). Staining of thecal plates revealed the plate formula typical for *Alexandrium* (Po, 4′, 6′′, 6c, 8(?)s, 5′′′, 2′′′′) (Figure 3). The first apical plate was slightly variable in its size and shape (see Appendix A), but generally short and consistently and entirely disconnected from the apical pore plate (Po). In most cases it was pentagonal with two anterior margins, with the left side touching plate 2′ being shorter than the right margin touching plate 4′ (Figure 3A,B). However, plate 1′ could also be rather quadrangular with just one long apical suture and without contact to plate 2′ (Figure 3C). A usually large ventral pore (vp) was located above plate 1′ at the junction of plates 1′, 2′ and 4′ and could occasionally also be seen in light microscopy (Figure 2C). When the left anterior margin of plate 1′ was missing the vp was located at the confluence of plates 1′, 2′, 4′ and 1′′. The vp may touch the 1′ plate but in many cases was slightly more anterior in position on the suture of plates 2′ and 4′ (Figure 3A,B). Exceptionally, no vp could be detected or two vp were present (Appendix A). The Po had a rounded dorsal and more pointed ventral side and had a large comma shaped pore (Figure 3D,E). The three apical plates surrounding Po were comparable in size with an almost symmetrical plate 3′ in dorsal position (Figure 3D). Precingular plates were of comparable size (Figure 3D), but the ventrally located plate 6′′ was distinctly smaller, comparable in size to plate 1′, pentagonal in shape, and longer than wide (Figure 3A–C). The anterior sulcal plate (sa) was located below plate 1′ and 6′′. It was very narrow and its left lateral suture to plate C1 was not extending the left lateral suture of plate 1′ with plate 1′′ (Figure 3A–C,G). The posterior sulcal plate (sp) was variable in shape and appearance (Appendix A), but generally elongated, longer than wide, and with a characteristically V-shaped anterior part touching the other sulcal plates (Figure 3B,G,H). This plate most often was rather smooth (Figure 3G,H), but also could have a straight or slightly curved line or groove eventually ending with a small pore (Figure 3I). In the central sulcal area six smaller plates could clearly be identified (Figure 3G). Plate ssa was large and appeared more as a precingular than a sulcal plate and had a small list around its sutures (Figure 3A–C,F,G). The right posterior sulcal plate (sdp) was slenderer and longer than the left posterior sulcal plate (ssp) (Figure 3G). The presence of two additional very tiny accessory sulcal plates was adumbrated but could not be unambiguously demonstrated.

The newly obtained AY7T large subunit (LSU) sequence was identical with most LSU reference sequences (Table 1). Only strain AY4T differed from all others (including the new AY7T) by two nucleotides. Three identical *A. taylorii* LSU sequences from Japan (strains Atay99Shio, Appendix A) revealed significant base pair differences compared to Mediterranean *A. taylorii,* e.g., 5.5% different to AY7T, 8.3% different to AY1T, or 8.1% different to AY4T. There were no previously deposited LSU sequences of strain AY7T.

Internal Transcribed Spacers (ITS) sequence comparison of AY7T revealed 100% identity with AY7T sequences previously deposited in Genbank and with most other ITS *A. taylorii* reference sequences. Among other Mediterranean strains, only strains CSIC-AV8 (La Fosca, Spain) and VGOE6 (Pagueroa, Spain) differed from other Mediterranean strains (including the new AY7T) by one nucleotide each (Table 2). However, as was the case for the LSU sequences, there were significant differences between ITS sequences of Mediterranean strains and ITS of the Japanese strain Atay99Shio-06 (AB841262.1, Appendix A), with, for example, 109 bp differences compared to strain AY7T (equivalent to 18.8%), or 107 bp differences compared to strain AT1T (equivalent to 21.6%), respectively.

### 2.2. Toxin Analysis

#### 2.2.1. PST

No PST were detected in *A. taylorii* AY7T by either ion pair chromatography coupled to post-column derivatization and fluorescence detection (LC-FLD) (Appendix A) or hydrophilic interaction liquid chromatography-tandem mass spectrometry (HILIC-MS/MS) (Appendix A), two independent methodological approaches. LC-FLD resulted in higher detection limits ranging for individual PST from 25 to 715 fg cell^−1^. In contrast, HILIC-MS/MS yielded orders of magnitude lower detection limit (LOD) between 0.1 and 1.9 fg cell^−1^ (Table 3).

#### 2.2.2. Lipophilic Compounds

In addition to PST, the *A. taylorii* AY7T strain also was analysed for other toxin groups known to be produced by species of the genus *Alexandrium*, namely cycloimines, such as spirolides (SPX), gymnodimines (GYM) and goniodomins (GD). No cycloimines were detected above the LOD of 0.6 fg cell^−1^ of SPX and 0.8 fg cell^−1^ of GYM based on the molecular response of 13-desmethyl SPX (SPX-1) and GYM A, respectively. *Alexandrium taylorii* was instead found to contain goniodomin A (GDA) (Figure 4) at a level of 11.7 pg cell^−1^. In addition, our data/analysis profiles/analytical results showed evidence of additional GD analogues that will be the subject to future research.

#### 2.2.3. Lytic Capacity

The dose response curve of *Rhodomonas* cell lysis exposed to different *A. taylorii* densities (Figure 5) revealed no significant effect on the target cells for the two lowest *A. taylorii* AY7T concentrations <50 cell mL^−1^. At higher *A. taylorii* densities, the number of intact *Rhodomonas* decreased consequently and total cell lysis was observed at the highest *A. taylorii* concentration of 1.9 × 10^3^ cells mL^−1^. EC_50_ was calculated as 185 cells mL^−1^ (95% confidence interval: 176–195 cells mL^−1^).

## 3. Discussion

Species-level toxinological data from the literature are only as good as the underlying taxonomical determination of the species/strains under study. It is therefore desirable to document the identification of the organism either morphologically or with molecular techniques when conducting chemical toxin analyses, as was done here. If the strain is sufficiently documented elsewhere, a strain identifier and literature citation should be provided. However, use of a previously described strain does not provide 100% certainty, as cross-contamination or even misidentification at the culture collection cannot be ruled out. The problem of reliable species identification especially refers to *Alexandrium* where most taxa are rather similar in general size and shape [21]. *Alexandrium* species identification is thus not a simple task and requires a thorough examination of subtle morphological differences in size and shape of diagnostic thecal plates such as the apical pore plate, the first apical plate, or of sulcal plates [3,21]. Moreover, recent phylogenetic studies revealed cryptic speciation and also invalidated some of the described morphospecies [30,31]. A prominent example of an *Alexandrium* morphospecies concept failure for species circumscriptions is the former *A. tamarense* species complex consisting of the morphospecies *A. tamarense*, *A. catenella*, and *A. fundyense* [30,32,33], where new species (i.e., *A. catenella*, *A. mediterraneum*, *A. tamarense*, *A. pacificum, A. australiense*) are now defined based on sequence data and the segregation into five genetic distinct clades [34].

Species determination of *A. taylorii* is also challenging. The description of the species [6] was based on field samples and no DNA sequences are available and linked to the type material. This ambiguity is illustrated by divergent sequences deposited in GenBank under the name *A. taylorii*; sequences labelled as *A. taylorii* based on strains isolated from Japan (Appendix A) differ substantially from sequence data obtained from Mediterranean *A. taylorii*. No morphological data are linked to the Japanese strains. For Mediterranean *A. taylorii*, field populations of two Spanish coastal sites were compared morphologically with *A. taylorii* populations from the type locality (French Atlantic) and were found to be within the range of intraspecific morphological variability [14]. For Mediterranean strain-based sequence data, there is thus at least indirect evidence that their morphology is likely to conform to *A. taylorii* sensu Balech. Moreover, for five Mediterranean strains, for which ITS sequence data were deposited in GenBank, morphology was examined by staining thecal plates [7], even though description or micrograph documentation was not provided. Likewise, six strains of *A. taylorii* from the Mediterranean with identical ITS sequences were examined morphologically by thecal plate dissection, and two *A. taylorii* cells of unknown strain identity were depicted [18]. For Adriatic *A. taylorii* strains with sequence data deposited at GenBank (AY1T, AY2T AY4T, AY7T, AY10T), no detailed morphological examination is published yet, but are now available for AY7T (Figure 2; Figure 3, Appendix A).

Morphology of AY7T largely conformed with the original species description of *A. taylorii* [6]. Cells of AY7T were slightly larger (length range 34.3–48.2 µm) than reported by Balech [6] (length range 31–44 µm), and larger than cells of the Spanish *A. taylorii* field population, where cell length ranged from 27 to 43 µm and from 26 to 43 µm for cells from Palmira and La Fosca, respectively [14]. In the original [6] and subsequent species descriptions [21], Balech did not explicitly mention variability in shape of plate 1′. However, such a variability, ranging from asymmetrical pentagonal to almost quadrangular, without contact to plate 2′ is evident in strain AY7T and has been documented for field population from the type locality (Arcachon, French Atlantic) and also from the Spanish Mediterranean [14]. Moreover, position of the ventral pore was also variable in the before mentioned field study [14] and in strain AY7T (Figure 3A–C, Appendix A). Notably, the exceptional presence of two ventral pores (Appendix A) was also noted by Balech [21] and Delgado et al. [14]. An exceptional lack of a ventral pore (Appendix A) was not reported before but confirms previous notions for other *Alexandrium* species that presence/absence of a vp is not a stable character [35]. One feature of thecal pattern differed consistently in strain AY7T compared to Balech’s original species description: the anterior sulcal plate sa. This plate is described and depicted by Balech [6,21] as very long with a significant anterior contact line to plate 1′′. However, for strain AY7T, sa was narrow and its right border was almost lined up with the right suture of plate 1′ so that there was almost no contact of plate sa and 1′′ (Figure 2A–C,F, Appendix A). Such a narrow sa plate was also described by Delgado et al. [14] for field samples from the type locality and from the Mediterranean and is also visible for cultured specimens from other Mediterranean localities [18,19,36]. Length of sa plate and the relative position of its right suture is thus not a constant and reliable feature of *A. taylorii*. In cultured cells of AY7T there was also a large variability in shapes of the posterior sulcal plate. Whereas this plate was consistently longer than wide and oblique to the right, there was consistent variability in presence/absence of the oblique groove extending from the right margin to the center and the presence/absence of a small pore at its end. Balech [21] noted that such a pore occurred “frequently” but it was only occasionally visible in cells of AY7T. Despite such minor deviation from Balech’s cell description, we are confident that strain AY7T corresponds to *A. taylorii*.

The notion that other Mediterranean *A. taylorii* strains do not produce PST [17,18,19] is now substantiated by detailed data on strain AY7T, which is solidly based on two different analytical methods and detailed estimated LOD values of about 1 fg cell^−1^. Given the fact that PST-producing *Alexandrium* species usually have PST cell quotas in the pg cell^−1^ range, *A. taylorii* AY7T can be regarded as a non-PST-producing strain. However, it has to be discussed whether or not toxin production is a stable species-specific trait. Whereas for a given clonal strain toxin production is generally proposed to be a genetically fixed and stable character [3,37], both toxic and non-toxic strains of the same species may occur. For PST, recent molecular work on presence/absence of genes responsible for toxin production as well as chemical toxin analysis of multiple strains indicate that, among the new species of the former *tamarense/catenella/fundyense* species complex, strains of *A. catenella* and *A. pacificum* consistently produce PST, whereas strains of *A. tamarense* and *A. mediterraneum* do not [32]. However, whereas most strains of the fifth species of this complex, *A. australiense*, do not produce saxitoxins above detection limits, one PST-producing strain of this species was described [38]. Likewise, for the very well-studied *A. minutum* and *A. ostenfeldii,* both PST-producing strains and strains without PST production have been reported [31,39,40,41]. Conflicting reports of PST-producing and non-toxic strains within less well studied *Alexandrium* are also present, for example, for *A. affine* [42,43], *A. andersonii* [43,44], or *A. leii* [42,45]. Thus, the debated question as to whether PST production is a stable species attribute has no clear answer which underlines the value of the present study, adding sound data to clarify the situation for *A. taylorii*. Nevertheless, additional analyses of multiple *A. taylorii* strains from different areas are needed to finally evaluate if lack of PST production for *A. taylorii* is a stable species-specific trait, especially since one deviating report on PST in *A. taylorii* exists. For a Malaysian strain of *A. taylorii* Lim et al. [9] reported the presence of PST. Whereas the documented morphological examination of the strain supports the species determination, neither a strain identifier nor sequence data of the strain in question were provided. Of more importance, however, is the fact that the reported toxin amounts were fairly low (<1 fmol cell^−1^), and that the reported *A. taylorii* PST profile exactly matched with the PST profile of a strain of *A. ostenfeldii* which was simultaneously studied [9]. Although such a 1:1 match of PST toxin profile of two different but simultaneously analysed *Alexandrium* species of course cannot be excluded, it may at least provoke some skepticism and the consideration of cross-contamination as a potential source of reporting the presence of trace PST amounts for *A. taylorii*. In any case, additional analyses of the Malaysian strain and other strains of *A. taylorii* from the Pacific area are urgently needed for a final clarification of the PST production potential of this species.

Whereas it is often stated that spiroimines within *Alexandrium* are only produced by *A. ostenfeldii* [3,46], corresponding analyses of these compounds for other *Alexandrium* species are largely missing. In general, reporting negative results is unspectacular and, to be ratable, require detailed reporting of the methods and limits of detection and quantification. Nevertheless, it is important to have this information for better understanding of the chemo-taxonomical relevance of toxins within *Alexandrium*. It is provided here with respect to excluding spirolides and gymnodimines from the toxin repertoire of Mediterranean *A. taylorii* AY7T.

The present results of lytic capacity of the Mediterranean AY7T confirm that *A. taylorii* produce and release lytic compounds. Another strain (AY1T) isolated from the same area has been shown before to negatively affect protistan target species [20]. Other reports on the presence of bioactive compounds and negative effects of Pacific *A. taylorii* on oyster larvae [8] or on *Artemia* and mammalian erythrocytes [10] are present in the literature. However, both papers do not provide supporting morphological and/or molecular evidence that indeed *A. taylorii* had been studied. In the paper of Emura et al. [10], not even a strain designation is included so that a reliable attribution of the reported finding to *A. taylorii* is considerably weakened.

Whereas lack of PST and presence of lytic activity of *A. taylorii* thus confirms previous reports [19,20], the presence of goniodomin A in *A. taylorii* as reported here has not been reported before. Goniodomin A production by *A. taylorii* might have been expected as this matches with its phylogenetic placement. In rRNA based phylogenetic trees *A. taylorii* forms a well-supported clade with *A. monilatum* (the type of Balech’s subgenus *Gessnerium*), *A. pseudogonyaulax*, *A. hiranoi* and *A. satoanum* [22,23]. Among those, GDA has been identified in *A. monilatum* [28], *A. pseudogonyaulax* [29], and *A. hiranoi* [27,47], while GDA production of *A. satoanum* has not yet been investigated. All species of this cluster belong to the subgenus *Gessnerium* which is defined by species where the first apical plate 1′ is not connected and not linked in any way with the apical pore plate [21]. However, some of the species that morphologically are classified into *Gessnerium*, and for which molecular data are available, such as *A. insuetum*, *A. margalefii* and *A. pohangense*, clearly cluster outside of the core *Gessnerium* group [3,48]. Thus, Balech’s morphological definition does not define a monophyletic group and new morphological unifiers for the core *Gessnerium* species would be needed. Anyhow, the current chemical evidence indicates that the *Gessnerium* core-species might chemotaxonomically be unique by presence of GDA and lack of PST, but this hypothesis will require the analysis of a higher number of *Alexandrium* species and strains, including *A. satoanum*, *A. margalefii*, *A. insuetum* and *A. pohhangense*, as well as yet uncultured species with a pentagonal and disconnected 1′ plate (*A. balechii*, *A. foedum*, *A. concavum*, *A. camurascutulum*, *A. globosum*) for the presence of GDA. Crude extract of the GDA producing *A. monilatum* was shown to cause hemolysis to erythrocytes from several mammalian species including humans [49], but lytic capacity of purified GDA has not yet unequivocally been shown. Future studies are needed to test if the lytic activity of AY7T (Figure 5) towards protistan targets are due to GDA or caused (or intensified) by other yet uncharacterized extracellular compounds.

Goniodomin A production by *A. taylorii* is of importance for the Mediterranean area where dense and recurrent blooms of this species occur [7,11,12,13,14,15,16]. For GDA producing *A. monilatum*, blooms have been linked to mortality of finfish and/or shellfish [50,51]. However, Mediterranean *A. taylorii* blooms have not yet been causatively linked to fish kills and are considered mainly to be of concern for tourism and recreational use of coastal waters and beaches [15,16]. Nevertheless, in 1999 a dense bloom with 27 × 10^6^ cells L^−1^ of *A. taylorii* in the lagoon of Marano (Northern Adriatic Sea) was associated with high mortality of seabass (*Dicentrarchus labrax*) which is extensively cultivated in the area (Beran et Cabrini, unpublished data; presented at the Riunione Scientifica Annuale del Gruppo di Algologia Italiana, Ancona, 2000). Thus, future studies of Mediterranean *A. taylorii* blooms should include analysis of GDA and should draw attention to potential links to fish kills or other environmental damage.

## 4. Materials and Methods

### 4.1. Strain Isolation and Harvest

Strain AY7T (=CoSMi1017) of *Alexandrium taylorii* was isolated from a benthic sample collected in the lagoon of Marano in May 2000. The lagoon of Marano is a shallow and semi open lagoon connected to the Northern Adriatic Sea. Salinity during summer normally ranges from 29 to 36. Part of the lagoon is divided in so-called “valli di pesca”, where seabass (*Dicentrarchus labrax*) is maintained in extensive culture. A massive bloom of *A. taylorii* identified by epifluorescence light microscopy using calcofluor [52] during July/August 1999 caused the loss of most of the stock. Standard tests for PST using HPLC in 1999 were negative and it was concluded at the time that the high fish mortality had probably been caused by occlusion of the gills, where many *A. taylorii* cells had been found in samples of dead fish. It was decided to test in 2000 if the mud would release fresh *A. taylorii* cells from resting cysts—in fact a second much smaller bloom developed in July 2000 without serious consequences.

For cell isolation, samples of ca 1 mL of mud were incubated in 50 mL of half strength medium B [53] (salinity 32) at 20 °C under cool white fluorescent light (80 μmol photons m^−2^ s^−1^) at a light: dark cycle of 12:12 h. First motile cells appeared in the sample after six days. Single cells were washed by transferring them three times into fresh medium under a dissection microscope (M10, Wild, Heerbrugg, Switzerland) using drawn micropipettes. Finally, single isolates were incubated at the same conditions into single wells of 24 well tissue culture plates (Corning, New York, NY, USA) containing 1 mL of medium. Growing cultures were adapted to full strength medium B. Several clonal strains were isolated but only strain AY7T was maintained and is now integrated in the Culture Collection of Sea Microorganisms (CoSMi) at the OGS—Trieste (http:/cosmi.inogs.it) as strain CoSMi1017.

For the experiment reported here, the strain was grown in a seawater K-medium [54] supplemented with selenite, prepared from 0.2 µm sterile-filtered (VacuCap, Pall Life Sciences, Dreieich, Germany) North Sea seawater (salinity of 32) at 15 °C, under cool-white fluorescent light at a photon flux density (PFD) of 50 μmol photons m^−2^ s^−1^ on a 16 h light: 8 h dark photo-cycle. For DNA sampling strain AY7T was grown in 70 mL plastic culture flasks. Cells in exponential phase were harvested by centrifugation at 3220× *g* for 10 min (Eppendorf 5810R, Hamburg, Germany) of 50 mL culture, and cell pellets were stored at −20 °C until further analysis. For toxin analysis, strain AY7T was grown in 250 mL plastic culture flasks under standard culture conditions. Cell concentrations from cultures in early stationary phase (at cell densities ranging from 1000 to 2000 cells mL^−1^) were determined by settling Lugol’s iodine-fixed samples and counting >400 cells under an inverted microscope. Cell pellets were harvested by centrifugation (Eppendorf 5810R, 3220× *g*, 10 min) and one pellet containing 227,000 cells was extracted for lipophilic toxins with 500 μL methanol, and another pellet containing 37,900 cells was extracted for paralytic shellfish toxins (PST) with 500 μL 0.03 M acetic acid, respectively. Therefore, samples were reciprocally shaken for 45 s at 6. 5 m s^−1^ with 0.9 g lysing matrix D (Thermo Savant, Illkirch, France) in a FP120 FastPrep instrument. Extracts were then centrifuged (Eppendorf 5415 R) for 15 min at 16,100× *g* at 4 °C. Each supernatant was transferred to a 0.45 µm pore-size spin-filter (Millipore Ultrafree), and centrifuged for 30 s at 800× *g*, the resulting filtrate being transferred into an ultra performance liquid chromatography (UPLC) autosampler vial for UPLC–MS/MS analysis.

### 4.2. Microscopy

Observation of living or fixed cells (formaldehyde: 1% final concentration, or neutral Lugol-fixed: 1% final concentration) was carried out using a compound microscope (Axiovert 2; Zeiss, Göttingen, Germany) equipped with epifluorescence and differential interference contrast optics. Light microscopic examination of thecal plates of *A. taylorii* was performed on fixed cells (neutral Lugol) stained with Solophenyl Flavine 7GFE500, a fluorescent dye specific to cellulose [55], which were examined with epifluorescence filter set 09 (Zeiss; BP 450-490; FT 510; LP 515). Images were taken with a digital camera (Axiocam MRc5; Zeiss). Cell length and width were measured at 1000× microscopic magnification using freshly fixed cells (formaldehyde, 1% final concentration) from dense, but healthy and growing strains (based on stereomicroscopic inspection of the living material) at early exponential phase and the Axiovision software (Zeiss).

### 4.3. DNA Extraction and Sequencing

For DNA extraction, the cell pellets were rinsed with 1 mL pre-heated (60 °C) PL1 DNA lysis buffer of the NucleoSpin Plant II DNA extraction kit (Macherey & Nagel, Düren, Germany). The lysis buffer containing the cells was subsequently transferred to a 2 mL cryovial prefilled with 200 µL glass beads (acid-washed, 212–300 µm, Sigma-Aldrich, St. Louis, MO, USA) and stored at −20 °C. DNA was extracted using the NucleoSpin Plant II kit according to the manufacturer’s instructions, with an additional cell disruption step within the beat tubes. Therefore, the samples were shaken for 45 s and another 30 s at a speed of 4.0 m s^−1^ in a cell disrupter (FastPrep FP120, Thermo-Savant). DNA elution was performed according to the manufacturer’s instructions, using 2 × 30 µL of the provided elution buffer, leading to a total elution volume of 60 µL.

The extracted DNA of *A. taylorii* AY7T was subjected to polymerase strain reaction (PCR), amplifying the large subunit (LSU/28S, D1-D2 region) and the Internal Transcribed Spacers (ITS1-5.8S-ITS2) of the ribosomal DNA (rDNA). The forward and reverse primers for amplification of 28S rDNA were Dir-F (5′-ACC CGC TGA ATT TAA GCA TA-3′) and Dir-2CR (5′-CCT TGG TCC GTG TTT CAA GA-3′), respectively. The primers for amplification of the ITS region were ITSa (5′-CCA AGC TTC TAG ATC GTA ACA AGG (ACT)TC CGT AGG T-3′) and ITSb (5′-CCT GCA GTC GAC A(GT)A TGC TTA A(AG)T TCA GC(AG) GG-3′), respectively. Each PCR reaction contained 16.3 μL of high-grade PCR H_2_O, 2.0 μL of Hotmaster Taq PCR Buffer (10×) (Quantabio, Beverly, MA, USA), 0.2 μL of each primer (10 μM), 0.2 μL of dNTP (10 μM) (Quantabio), 0.1 μL of Taq Polymerase (Quantabio) and 1 μL of DNA template (10 ng μL^−1^) to a final volume of 20 μL.

Cycler conditions for LSU amplification were as follows: initial denaturation at 94 °C for 2 min, followed by 30 cycles of denaturation at 94 °C for 30 s, annealing at 55 °C for 30 s and elongation at 68 °C for 2 min. A final extension step at 68 °C for 10 min was performed. Cycler conditions for ITS amplification were as follows: initial denaturation at 94 °C for 4 min, followed by 10 cycles of denaturation at 94 °C for 50 s, annealing at 58 °C for 40 s and elongation at 70 °C for 1 min, then 30 cycles of denaturation at 94 °C for 45 s, annealing at 50 °C for 45 s and elongation at 70 °C for 1 min. A final extension step at 70 °C for 5 min was performed.

The PCR amplicons were run on a 1% agarose gel at 70 mV for 40 min in TE buffer to verify that the PCR amplicons were of the expected length. The PCR amplicon was purified using the NucleoSpin Gel and PCR clean-up kit (Macherey-Nagel, Düren, Germany) and sequenced directly in both directions on an ABI PRISM 3730XL (Applied Biosystems by Thermo Fisher Scientific, Waltham, MA, USA) as described in Tillmann et al. [56]. Raw sequence data were processed using the CLC Genomics Workbench 12 (Qiagen, Hilden, Germany).

Gained LSU and ITS sequences of the actual sample of *A. taylorii* AY7T were aligned and compared to published sequences of *A. taylorii* (Appendix A) using the MUSCLE algorithm implemented in the software MEGA7 (version 7.0.26; [57]). ITS sequence data of strain AY7T previously (2006) deposited in GenBank (Acc no. AM296012.1) were included in the comparison.

### 4.4. Toxin Analysis

A cell pellet was extracted with 300 µL 0.03 M acetic acid and another with 300 µL methanol for lipophilic toxins and lyzing Matrix D (Thermo Savant) in a homogenizer (MagnaLyzer, Roche Diagnostics, Mannheim, Germany) for 45 s at 5500 m s^−1^. The homogenates were centrifuged for five min at 13,200× *g*. The supernatants were transferred to spin filters (0.45 µm, UltraFree, Millipore, Eschborn, Germany) and centrifuged for 30 s at 5700× *g*. The filtrates were transferred to HPLC vials and stored at −20 °C until analysis.

#### 4.4.1. Paralytic Shellfish Toxins

PSP toxin (PST) analysis was performed by two independent methodological approaches: by ion pair chromatography coupled to post-column derivatization and fluorescence detection (LC-FLD) and hydrophilic interaction liquid chromatography coupled to tandem mass spectrometry (HILIC-MS/MS).

LC-FLD analysis was performed on a LC1100 series liquid chromatography system consisting of a G1379A degasser, a G1311A quaternary pump, a G1229A autosampler, and a G1321A fluorescence detector (Agilent Technologies, Waldbronn, Germany), equipped with a Phenomenex Luna C18 reversed-phase column (250 mm × 4.6 mm id, 5 µm pore size) (Phenomenex, Aschaffenburg, Germany) with a Phenomenex SecuriGuard precolumn. The column was coupled to a PCX 2500 post-column derivatization system (Pickering Laboratories, Mountain View, CA, USA). Eluent A contained 6 mM octane-sulfonic acid, 6 mM heptane-sulfonic acid, 40 mM ammonium phosphate, adjusted to pH 6.95 with dilute phosphoric acid, and 0.75% tetrahydrofuran. Eluent B contained 13 mM octane-sulfonic acid, 50 mM phosphoric acid, adjusted to pH 6.9 with ammonium hydroxide, 15% acetonitrile and 1.5% tetrahydrofuran. The flow rate was 1 mL min^−1^ with the following gradient: 0–5 min isocratic A, 15–16 min switch to B, 16–35 min isocratic B, 35–36 min switch to A, 36–45 min isocratic A. The injection volume was 20 µL and the autosampler was cooled to 4 °C. The eluate from the column was oxidized with 10 mM periodic acid in 555 mM ammonium hydroxide before entering the 50 °C reaction coil, after which it was acidified with 0.75 M nitric acid. Both the oxidizing and acidifying reagents entered the system at a rate of 0.4 mL min^−1^. The toxins were detected by dual-monochromator fluorescence (l_ex_ 333 nm; l_em_ 395 nm). The data were processed with Chemstation software (Agilent, Santa Clara, CA, USA) and calibrated against external standards.

HILIC-MS/MS analysis was achieved on an Acquity UPLC Glycan BEH Amide column (130 Å, 150 mm × 2.1 mm, 1.7 µm, Waters, Eschborn, Germany) equipped with an in-line 0.2 µm Acquity filter and thermostated at 60 °C with an isocratic elution to 5 min with 98% eluent B followed by a linear gradient of 2.5 min to 50% B and 1.5 min isocratic elution. The flow rate was 0.4 mL min^−1^, and the injection volume was 2 µL. Mobile phase A consisted of water with 0.15% formic acid and 0.6% ammonia (25%). Mobile phase B consisted of water/acetonitrile (3:7, *v*/*v*) with 0.1% formic acid. Mass spectrometric experiments were performed in the selected reaction monitoring (SRM) mode on a Xevo TQ-XS triple quadrupole mass spectrometer equipped with a Z-Spray source (Waters, Halethorpe, MD, USA). Instrument parameters are given in Appendix A and used mass transitions in Table 4. PSTs were quantified by external calibration with standard mix solutions of 4 concentration levels consisting of the following PSTs: STX, NEO, GTX2/3, GTX1/4, dcSTX, dcGTX2/3, B1, and C1/2. All individual standard solutions were purchased from the Certified Reference Materials Program (CRMP) of the Institute for Marine Biosciences, National Research Council (Halifax, Canada).

#### 4.4.2. Lipophilic Compounds

LC-MS/MS analysis for lipophilic toxins was performed on a reversed phase C18 column (Purospher STAR RP-18 end-capped (2 µm) Hibar HR 50-2.1, Merck, Darmstadt, Germany) equipped with a guard column (EXP Pre-column Filter Cartridge, Merck) and thermostated at 40 °C with an isocratic elution to 5 min with 5% eluent B followed by a linear gradient of 2.0 min to 100% B and 3.0 min isocratic elution prior to return to initial conditions. The flow rate was 0.6 mL min^−1^, and the injection volume was 0.5 µL. Mobile phase A consisted of 500 mL water with 955 µL formic acid and 75 µL 25% ammonia. Mobile phase B consisted of 475 mL acetonitrile, 25 mL deionized water, 955 µL formic acid and 75 µL 25% ammonia. Mass spectrometric experiments were performed in the selected reaction monitoring (SRM) mode in positive polarity on a Xevo TQ-XS triple quadrupole mass spectrometer equipped with a Z-Spray source (Waters). Instrument parameters are given in Appendix A and used mass transitions in Table 5. A standard solution of 500 pg µL^−1^ GDA [58] was used for quantification. Standard solutions of 100 pg µL^−1^ SPX 1 and 50 pg µL^−1^ GYM A (CRMP, IMB-NRC, Halifax, NS, Canada) were used for the determination of detection limits.

#### 4.4.3. Lytic Compounds

The presence of extracellular bioactive compounds with lytic capacity was investigated using a whole cell cryptophyte *Rhodomonas salina* 24-h-bioassay [59,60]. *Rhodomonas salina* (Strain KAC30) was grown with the same medium and light/temperature settings as described for *A. taylorii*. A culture of *A. taylorii* AY7T at late exponential phase (1.9 × 10^3^ cells mL^−1^) was used to prepare triplicate glass-vials (3.9 mL each) with seven dilutions spanning from 0.02 × 10^3^ to 1.9 × 10^3^ cells mL^−1^. Triplicate glass-vials with culture medium served as control. A dense *R. salina* culture was diluted with filtered culture medium to a density of 4 × 10^5^ cells mL^−1^. Each sample including controls was spiked (100 µL) with this *R. salina* culture to yield a final *R. salina* concentration of 1 × 10^4^ cells mL^−1^ and a final assay volume of 4 mL Samples were incubated for 24 h in the dark at 15 °C. Subsequently, samples were fixed with Lugol’s iodine solution (2% final conc.) and intact target cells were counted with an inverted microscope (Axiovert 40c, Zeiss). Percentage of intact *Rhodomonas* cells were calculated as Rho_final_/Rho_control_ × 100%. EC_50_ was calculated using the non-linear fit procedure of Statistika (version 9.1, StatSoft, Tulsa, OK, USA) regression of a sigmoidal curve as %intact cells = 100/[1 + (X/EC_50_)^h^]; with X = the log-transformed *A. taylorii* cell concentrations and EC_50_ and h as fit-parameters. EC_50_, i.e., the concentration of *A. taylorii* where 50% of *Rhodomonas* were lysed, is expressed as cells mL^−1^, including 95% confidence intervals.

## Figures and Tables

**Figure 1 toxins-12-00564-f001:**
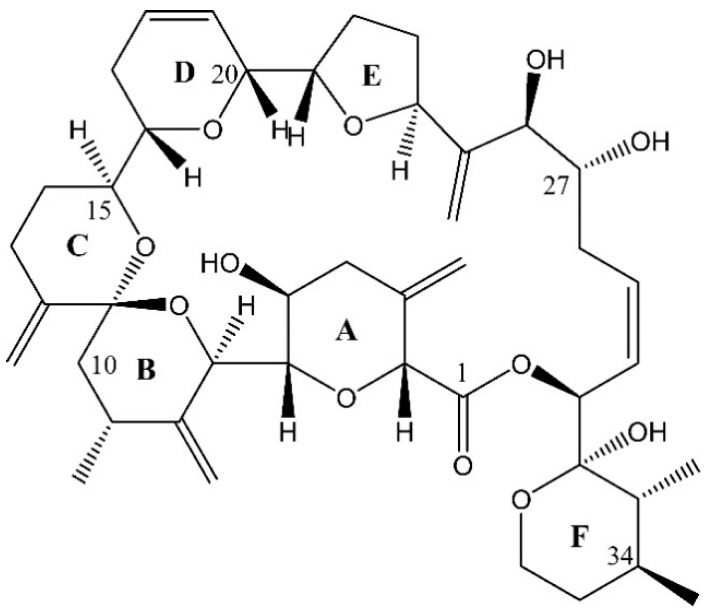
Chemical structure of goniodomin A.

**Figure 2 toxins-12-00564-f002:**
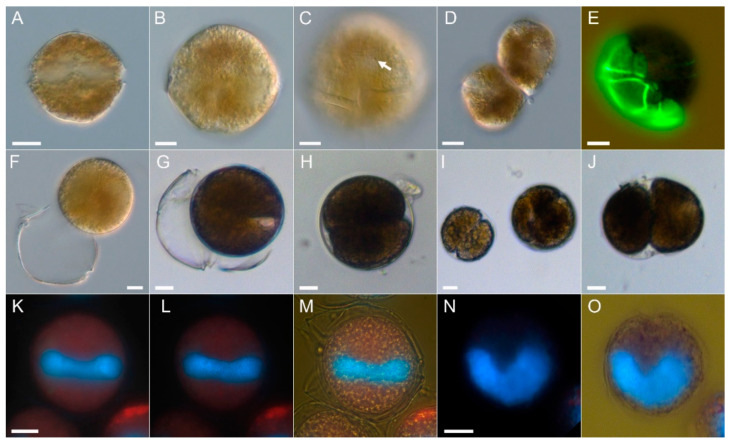
*Alexandrium taylorii* AY7T, LM micrographs of living (**A**–**D**,**F**–**J**) or fixed (**E**,**K**–**O**) cells. (**A**–**C**) General size and shape. Note the ventral pore (arrow) in (**C**). (**D**) Newly divided motile pair of cells. (**E**) Newly divided cell stained with Solophenyl Flavine showing presence of half of the parent thecal plates. (**F**,**G**) Temporary cyst formation after ecdysis of the whole theca. (**H**–**J**) Different temporary cysts with cells in division. (**K**–**M**) Different focal planes and illumination of the same cell stained with DAPI to indicate shape and position of the nucleus (blue). (**N**,**O**) Two views of the same DAPI-stained cell in apical view. Scale bars = 10 µm.

**Figure 3 toxins-12-00564-f003:**
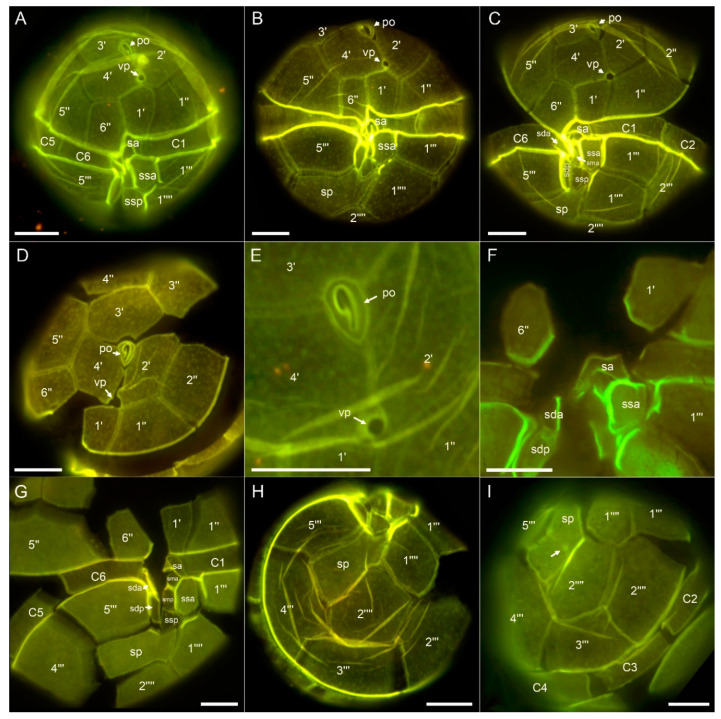
*Alexandrium taylorii* AY7T, different thecae of Lugol-fixed cells stained with Solophenyl Flavine and viewed with epifluorescence and blue light excitation. (**A**–**C**) Cells in ventral view. (**D**) Epithecal plates in apical view. (**E**) Detailed apical view of the pore plate (Po) and the ventral pore (vp). (**F**) Detailed view of the sulcal area to show shape of the anterior sulcal plate (sa). (**G**) Hypothecal and sulcal plates in ventral view. (**H**,**I**) Hypothecal plates in antapical view. Note the groove ending with a small pore (arrow in I). Plate labels according to the Kofoidian system. Sulcal plate labels: sp = posterior sulcal plate; sdp = right posterior sulcal plate; ssp = left posterior sulcal plate; sda = right anterior sulcal plate; smp = median posterior sulcal plate; sma = median anterior sulcal plate; sa = anterior sulcal plate. Scale bars = 10 µm.

**Figure 4 toxins-12-00564-f004:**
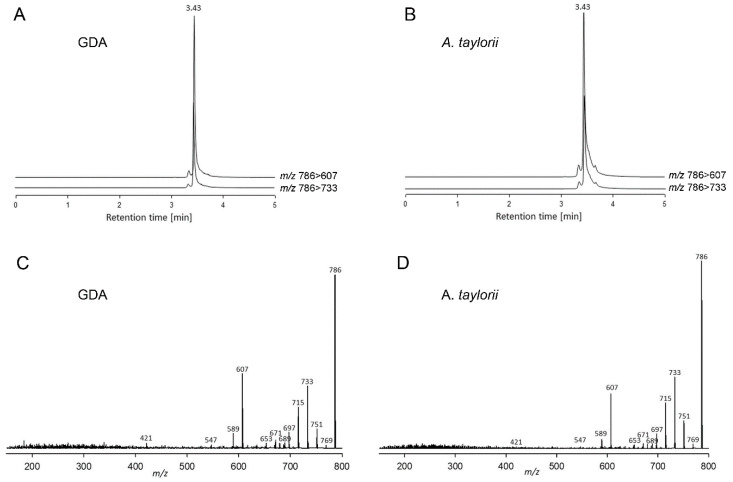
LC-MS/MS chromatograms of the ion transitions *m*/*z* 786 > 607 and 786 > 733 of (**A**) a goniodomin A (GDA) standard solution of 500 pg µL^−1^ and (**B**) a methanolic extract of *Alexandrium taylorii* AY7T as well as collision induced dissociation (CID) spectra of (**C**) a GDA standard solution of 500 pg µL^−1^ and (**D**) a methanolic extract of *Alexandrium taylorii* AY7T.

**Figure 5 toxins-12-00564-f005:**
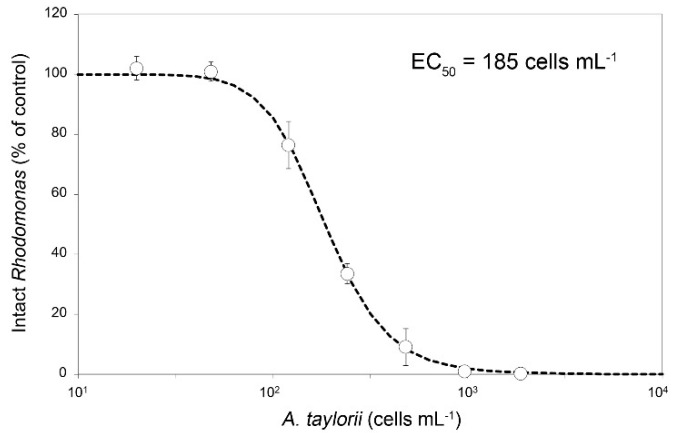
Cell bioassay with the cryptophyte *Rhodomonas salina* undergoing cell lysis when exposed to whole cells of *Alexandrium taylorii* AY7T. Intact target cells (% of control) plotted against log-transformed *A. taylorii* density (mL^−1^). Results are expressed as triplicate mean ± 1 SD.

**Table 1 toxins-12-00564-t001:** Sequence alignment of the homologous fragment for the *A. taylorii* large subunit (LSU) sequences.

Strain	Sequence (5′-3′)	Nucleotide Positions (bp)
AY7T (this study)	CATTAATTTGGACTTGGTGCAA	547–568
AY4T	---------------------AG---------------------	555–576
AY1T	----------------------------------------------	516–537
AY2T	----------------------------------------------	551–572

**Table 2 toxins-12-00564-t002:** Sequence alignment of the homologous fragment for the *A. taylorii* Internal Transcribed Spacers (ITS) sequences. Dots represent 134 base pairs, which are identical in the ITS sequences of all strains shown in the table.

Strain	Sequence (5′-3′)	Nucleotide Positions (bp)
AY7T (this study)	GATCCAA……….AGGCATC	354–360……….494–500
CSIC-AV8	------T------………..--------------	314–320……….454–460
VGO704, VGOE6	--------------………..------A-----	314–320……….454–460
AY10T	--------------………..--------------	314–320……….454–460
AY1T	--------------………..--------------	314–320……….454–460
AY7T	--------------………..--------------	314–320……….454–460
CBA-1	--------------………..--------------	314–320……….454–460
CNR-AT4	--------------………..--------------	314–320……….454–460
CNR-ATAYB2	--------------………..--------------	314–320……….454–460
Field sample	--------------………..--------------	314–320……….454–460
Temporary-cyst	--------------………..--------------	314–320……….454–460
VGO705	--------------………..--------------	314–320……….454–460

**Table 3 toxins-12-00564-t003:** Cellular detection limits (LOD) of paralytic shellfish toxins (PST) determined by ion pair chromatography coupled to post-column derivatization and fluorescence detection (LC-FLD) and hydrophilic interaction liquid chromatography-tandem mass spectrometry (HILIC-MS/MS). nd = not determined.

Toxin	LOD (FLD) [fg cell^−1^]	LOD (MS/MS) [fg cell^−1^]
C1	57	0.25
C2	57	0.76
C3	nd	0.49
C4	nd	1.87
B1	141	0.12
B2	nd	0.49
STX	35	0.13
NEO	516	0.63
GTX1	715	0.07
GTX2	26	0.24
GTX3	32	0.18
GTX4	722	0.16
dcSTX	51	0.15
dcNEO	nd	0.34
dcGTX1	nd	0.35
dcGTX2	25	0.49
dcGTX3	25	0.45
dcGTX4	nd	0.84
doSTX	nd	0.09
TTX	nd	0.18

**Table 4 toxins-12-00564-t004:** Mass transitions of PST and GC toxins. +/- indicates positive or negative ionization mode.

	Quantifier +	Qualifier +	Quantifier -	Qualifier -
doSTX	241 > 60	241 > 206		
dcSTX	257 > 126	257 > 222		
dcNEO	273 > 126	273 > 225		
STX	300 > 126	300 > 204		
NEO	316 > 126	316 > 220		
TTX	320 > 302	320 > 162		
dcGTX2			351 > 164	351 > 333
dcGTX3	353 > 255			351 > 333
dcGTX1			367 > 274	367 > 349
dcGTX4	369 > 271			367 > 349
B1	380 > 300			378 > 122
B2	396 > 316			394 > 122
GTX2			394 > 351	394 > 333
GTX3		396 > 298	394 > 333	
GTX1			410 > 367	410 > 349
GTX4		412 > 314	410 > 367	
C1			474 > 122	474 > 351
C2	396 > 298			474 > 122
C3		412 > 332	490 > 410	
C4	412 > 314			490 > 392
GC3	377 > 359	377 > 257		
GC3a	393 > 375	393 > 257		
GC6	393 > 375	393 > 273		
GC6a	409 > 391	409 > 273		
GC3b	457 > 359	457 > 377		
GC1/2	473 > 375	473 > 455		
GC6b	473 > 375	473 > 393		
GC1a/2a	489 > 409	489 > 471		
GC4/5	489 > 489	489 > 471		
GC4a/5a	505 > 425	505 > 487		
GC1b/GC2b	553 > 393	553 > 473		
GC4b	569 > 489	569 > 409		
GC5b	569 > 409	569 > 489		

**Table 5 toxins-12-00564-t005:** Mass transitions of monitored lipophilic toxins. + indicates positive ionization mode.

Toxin	Quantifier +	Qualifier +
GYM A	508 > 490	508 > 162
GYM D	510 > 492	
12-me-GYM A	522 > 504	
GYM B/C	524 > 506	
GYM E	526 > 508	
GYM (uncharacterized)	540 > 522	
GYM (uncharacterized)	542 > 524	
SPX (uncharacterized)	592 > 164	
SPX (uncharacterized)	618 > 164	
SPX H	650 > 164	
SPX I	652 > 164	
SPX (uncharacterized)	658 > 164	
SPX (uncharacterized)	666 > 164	
SPX (uncharacterized)	666 > 180	
SPX (uncharacterized)	678 > 150	
13,19-Didesmethyl-SPX C	678 > 164	
SPX (uncharacterized)	686 > 164	
SPX A	692 > 150	
13-Desme-SPX C, SPX G	692 > 164	
SPX (uncharacterized)	692 > 180	
SPX B	694 > 150	
13-Desme-SPX D, PnTx G, 20-Hydroxy-13,19-didesmethyl SPX C	694 > 164	
27-Hydroxy-13-desmethyl SPX C	694 > 180	
20-Hydroxy-13,19-didesmethyl-SPX D	696 > 164	
SPX (uncharacterized)	698 > 164	
SPX (uncharacterized)	704 > 164	
SPX (uncharacterized)	706 > 150	
SPX C, 20-Methyl-SPX G	706 > 164	
SPX D	708 > 164	
SPX (uncharacterized)	708 > 180	
SPX (uncharacterized)	710 > 150	
SPX (uncharacterized)	710 > 164	
SPX (uncharacterized)	718 > 164	
SPX (uncharacterized)	720 > 150	
SPX (uncharacterized)	720 > 164	
SPX (uncharacterized)	722 > 164	
SPX (uncharacterized)	722 > 180	
PnTx F	766 > 164	
PnTx E	784 > 164	
GDA	786 > 607	786 > 733

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
