# Peer review of "A Mediterranean Alexandrium taylorii (Dinophyceae) Strain Produces Goniodomin A and Lytic Compounds but Not Paralytic Shellfish Toxins"

_toxins, 2020, doi:10.3390/toxins12090564_

Round 1

Reviewer 1 Report

The paper entitled “Mediterranean Alexandrium taylorii (Dinophyceae) produce goniodomin A and lytic compounds but not paralytic shellfish toxins” reports the characterization of the toxic profile of this species which forms dense blooms but whose toxicity characterization presents some uncertainties. Alexandrium is one of the most studied dinoflagellate genus and many species produce very dangerous toxins, among which the best characterized are saxitoxins; goniodomins were more recently characterized and were found to cause fish kills so that it is important to know which species can produce them, especially if they are able to cause intensive blooms. In this paper the identification of the species is also accurately performed.

The manuscript is well organized, the experiments reported are well planned and conducted, results are consistent; thus the paper is worth of publication although with minor revisions. Specific comments are listed below where you can find that my major requirements are related to some methods specification and a main concern is related to a discussion statement.

Title

Maybe better: “A Mediterranean Alexandrium taylorii (Dinophyceae) strain produces goniodomin A and lytic compounds but not paralytic shellfish toxins”

Abstract

Line 20: change “gonidomin” to “goniodomin”

Introduction

Line 35: change “gonyaulacoide” to “gonyaulacoid”

Line 45: change to: a severe Alexandrium catenella bloom of outstanding….

Line 47-48: change to: While toxin production is well studied for the main PST-producing species, for example for species of the former tamarense/fundyense/catenella species complex or for A. minutum…

Line 57: change to: Mediterranean strains are usually listed and cited as non-PST-producers, however, this belief is not based on actual data or is simply based on “pers. comm.” information

Line 66-67: for which sequence data are available in GenBank, was shown to immobilize and lyse a protistan grazer which is indicative for production of extracellular lytic compounds

Results

Line 94: change “ways” with “types”

Lines 147-148: I am not sure the sentence is necessary as this is a newly set up culture and the table clearly states “this study”.

Line 168: change “the A. taylorii AY7T sample” to “A. taylorii AY7T strain”

Lines 172-174: Maybe better to write: A. taylorii was instead found to contain goniodomin A (GDA) (Figure 3) at a level of 11.7 pg cell-1. In addition, our data/analysis profiles/analytical results showed evidence of additional GD analogues that will be the subject to future research.

Line 179: Change: “The dose response curve of Rhodomonas cell lysis at different A. taylorii concentrations” to “The dose response curve of Rhodomonas cell lysis exposed to different A. taylorii densities”

Line 181: change rapidly with parallely/consequentely

Discussion

Lines 189-196: These sentences are not clear.

Line 197: add “genus” after Alexandrium.

Line 206: rewrite the sentence, it is too much schematic.

Lines 213-216: change to: “For Mediterranean strain-based sequence data, there is at least indirect evidence that their morphology likely conforms to A. taylorii sensu Balech”

Line 216: change “at GenBank” with “in GenBank” and “although” with “even though”

Lines 217-222: English is not sufficiently clear.

Line 236: insert “pattern” between “thecal” and “differed”.

Line 242: change to: Length of sa plate…

Line 246: insert “and” between “presence” and “absence”

Line 266: “data situation” here is not correct, write it differently, for example “the debated question if PST…

The following part of the discussion (lines 250-281) in my opinion poses the biggest concern. I think that this paper can clearly affirm that A. taylorii can produce Goniodomins but the results are not sufficient to state that this species does never produce SXTs. The importance you also give to your results is testified by the title (first is a goniodomin and lytic compounds producer, then a non-SXT producer). The first sentence (lines 250-252) is especially misleading. There are also important examples, such as the one concerning A. ostenfeldii which has strains producer and others not producer of SXTs as you also further discuss. I also think that it is not correct to base your statement on the analysis of only one strain which was isolated in a peculiar site, i.e. a lagoon, and also on the consideration that a previous paper was not correctly conducted; maybe you can discuss the different results on the base of different methods or of methods with different sensitivity or precision. Please rewrite the part of the discussion concerning toxicity findings in A. taylorii.

Lines 285-288: the sentence is not clear.

Line 299: change “production of” with “production by”; same in line 320.

Line 311: delete “their”; change to: “will require the analysis of a higher number of…

Material and Methods

Please add details (here or in the result section) on cyst morphology, e.g. how you identified them, if it was easy as they were conforming to previous description or if you could not immediately identify them as they had not been precisely described before so that you rather observed and identified the excysted cell morphology.

Report some characteristic of the Lagoon, e.g. salinity values (constant or not), depth, if it is connected to the Adriatic sea with major exchange or not, etc. In the introduction you could also add the part concerning A. taylorii bloom in the lagoon which is cited in the discussion, it would make more clear why you were looking for the cysts.

Report incubation conditions for excystment (temperature, salinity).  Apparently you obtained more than one strain, write if AY7T was  the only surviving strain.

Line 348: delete “of”

Line 415: write “shellfish” correctly

Line 467: write “with seven dilutions”

Refer the culture conditions of R. salina and, if highly different from those of A. taylorii, how you managed the differences, were the cells centrifuged before the spike addition? How much was the volume of the spike? How much was the total volume of the assay?

Fig. 3: change “an metabolic extract” to “a metabolic extract”

Fig. 4: Maybe better to write: “Cell bioassay with the cryptophyte Rhodomonas salina undergoing lytic activity when exposed to whole cells (or cultured cells) of Alexandrium taylorii AY7T.”

Reviewer 2 Report

This study reports morphological, molecular and toxicological data on a Mediterranean Alexandrium taylorii strain. This is one of the few studies reporting this kind of information on that species.

I found the paper very interesting. Information obtained adds important knowledge not only for A. taylorii but in general it contributes to the advancement of knowledge on the whole Alexandrium genus. I recommend the publication after minor revisions.

Title: I think that it should be: “…..produces goniodomin A….”

Lines 30-34 Toxic blooms are not always linked to high cell densities (i.e. Dinophysis). Please rephrase and use for example ‘Harmful Algal Bloom’. This concept includes the harmful events but it does not link only to blooms with high biomass.

Line 35 gonyaulacoide shoul be gonyaulacoid.

Line 52 ‘Alexandrium taylorii is a high biomass producer species…’

Line 236 ‘…consistently in strain AY7T compared to Balech’s…’.

Line 333 I understand that the strain was obtained from sediment incubation experiments not by isolation of single cyst. Please specify better.

Do you have an idea of the morphology of the cysts?

Line 353 ‘one pellets…and another pellet’ I think it should be better to include a volume of harvested culture.

Figure captions:

Figure 1. Line 104: check brackets

Figure 2. pore plate (po). Please use Po as in the text.

Reviewer 3 Report

In this manuscript, the authors found goniodomin A, cyclic polyethermacrolide, in Mediterranean Alexandrium gylorri, for the first time. PST was not identified in this Alexandrium species by LC-FLD and LC-MS/MS. The authors also confirm the lytic activity of this species, although causative compound was not identified. These finding are interesting. However, there is some major points that should be revised before publishing.

1) The structure of goniodomin A should be shown in introduction or result section.

2)Figure 3 and 4 should be shown more clearer. They should be shown sharply. It is hard to see details of the figures now.

3) Goniodomin A in this A. gylorri should be identified using high resolution mass spectrometer. The mass spectrum of goniodomin A its self from A. gylorri should be shown together with LCMSMS in Figure 3, because this is the major finding in this manuscript.  And 

4) Please show the photo of lytic Rhodomonas cell affected by A. taylorii.

5) The result of the test of lytic activity of authentic goniodomin A should be also shown in Figure 4 for comparison, to discuss the lytic activity of A. taylorii is related to goniodomin A or not.

6) Please add LCMS result that shows the absence of PST in SI, by comparison with a LCMS chart of authentic PST. 

Minor:

Line 20 “gonidomin” should be corrected to “goniodomin”. Please check typing mistakes in whole manuscript again.

Round 2

Reviewer 3 Report

1) I still think LCLFD and LCMS chromatograms of authentic PST of the amount close to LOD, and also these chromatograms of the sample, are needed to be shown in SI, because there is no evidence to prove that authentic PST at low concentration around LOD was accurately measured to mention that the amount of PST in the sample was less than LOD. This is important point for this manuscript to let the readers convinced that this strain did not contain PST. If the authors do not show such data, there is no evidence to prove this result. Absence of PST is a major finding of this manuscript. The result should be shown with evidence.

2) In Figure 4A and B, 876/607 and 876/733 on the chromatograms are mistakes?

They should be 786/607 and 786/733.

In addition, the applied amounts of authentic GDA for Figure 4A and C are needed to be described in the caption of Figure 4.

Author Response

Reviewer

1) I still think LCLFD and LCMS chromatograms of authentic PST of the amount close to LOD, and also these chromatograms of the sample, are needed to be shown in SI, because there is no evidence to prove that authentic PST at low concentration around LOD was accurately measured to mention that the amount of PST in the sample was less than LOD. This is important point for this manuscript to let the readers convinced that this strain did not contain PST. If the authors do not show such data, there is no evidence to prove this result. Absence of PST is a major finding of this manuscript. The result should be shown with evidence.

Reply: LC-FLD chromatograms and TIC chromatograms of PST standard mixes and the A. taylorii sample are provided as SI figures (Fig. S3 and S4).

2) In Figure 4A and B, 876/607 and 876/733 on the chromatograms are mistakes?

They should be 786/607 and 786/733.

Reply: this observation is correct and the mass transitions were corrected as suggested.

In addition, the applied amounts of authentic GDA for Figure 4A and C are needed to be described in the caption of Figure 4.

Reply: The concentration of the used GDA standard solution was 500 pg µL-1. This information was added to method section and the caption of Figure 4.

Round 3

Reviewer 3 Report

The manuscript has been well revised as requested. I think that now this manuscript is suitable for publication.